# FEDERATED LEARNING WITH $L_0$ CONSTRAINT VIA PROBABILISTIC GATES FOR SPARSITY

## ABSTRACT

Federated Learning (FL) is a distributed machine learning setting that requires multiple clients to collaborate on training a model while maintaining data privacy. The unaddressed inherent sparsity in data and models often results in overly dense models and poor generalizability under data and client participation heterogeneity. We propose FL with an $L_0$ constraint on the density of non-zero parameters, achieved through a reparameterization using probabilistic gates and their continuous relaxation: originally proposed for sparsity in centralized machine learning. We show that the objective for $L_0$ constrained stochastic minimization naturally arises from an entropy maximization problem of the stochastic gates and propose an algorithm based on federated stochastic gradient descent for distributed learning. We demonstrate that the target density ($\rho$) of parameters can be achieved in FL, under data and client participation heterogeneity, with minimal loss in statistical performance for linear and non-linear models: *(i)* Linear regression (LR). *(ii)* Logistic regression (LG). *(iii)* Softmax multi-class classification (MC). *(iv)* Multi-label classification with logistic units (MLC). *(v)* Convolution Neural Network (CNN) for multi-class classification (MC). We compare the results with a magnitude pruning-based thresholding algorithm for sparsity in FL. Experiments on synthetic data with target density down to $\rho = 0.05$ and publicly available RCV1, MNIST, and EMNIST datasets with target density down to $\rho = 0.005$ demonstrate that our approach is communication-efficient and consistently better in statistical performance.

## 1 INTRODUCTION

FL training algorithms are defined by the requirements of data privacy and the distributed nature of learning algorithms (McMahan et al., 2017). The optimization in FL is challenging due to data heterogeneity and the availability of clients or devices, making the averaging of models or gradients inefficient. FL systems also eliminate the need for centralization of data, even in cases where there is no privacy concern (Kairouz et al., 2021). The resources available at devices participating in FL vary across different settings. In cross-device FL, where edge devices are often hardware-restricted, reducing computational and communication overheads, either in training or inference, is beneficial (Wang et al., 2021). A sparsity-inducing learning methodology is desirable to meet the system requirements. A sparse model, as opposed to an overly trained dense model, is favored in machine learning (ML) for its generalizability (Tibshirani, 1996). Previous works include sparse regression (Bertsimas et al., 2020), and different algorithms on Lasso sparse regression (Frandi et al., 2016; Šehić et al., 2022).

The standard approach for achieving sparsity is to utilize the $L_p$ norm for regularization. The traditional Ridge ($L_2$) and Lasso ($L_1$) penalties depend on the magnitude of the weights, resulting in varying levels of shrinkage. In contrast, the $L_0$ regularizer has a constant penalty for non-zero parameters, making it a magnitude-independent penalizer. For this reason, we prefer to apply an $L_0$ constraint in FL to learn a global model with the desired parameter density ($\rho$). Consider $C$ clients in FL holding data $(D^{(c)})_{c=1}^{C} : (X^c, Y^c)$ where $X^c \in \mathbb{R}^{n_c \times p}$, $Y^c \in \mathbb{R}^{n_c}$ and $\sum_{c=1}^{C} n_c = N$. Assuming a linear model $h(x, \theta) : \mathbb{R}^p \to \mathbb{R}$ and loss $\ell(h(x; \theta), y)$ where $x \in \mathbb{R}^p, y \in \mathbb{R}$, and $\theta \in \mathbb{R}^p$,

the $L_0$ density constrained minimization problem and the associated Lagrangian can be written as

$$\min_\theta \quad \sum_{c=1}^{C} \frac{n_c}{N} \mathcal{L}^{(c)}(\theta) \quad \text{subject to} \quad \frac{\|\theta\|_0}{|\theta|} \leq \rho, \quad \|\theta\|_0 = \sum_{j=1}^{|\theta|} \mathbb{I}[\theta_j \neq 0], \text{ and} \tag{1}$$

$$\mathfrak{L}(\theta, \lambda) = \sum_{c=1}^{C} \frac{n_c}{N} \mathcal{L}^{(c)}(\theta) + \lambda \left( \frac{\|\theta\|_0}{|\theta|} - \rho \right). \tag{2}$$

Here, $\mathcal{L}^{(c)}(\theta)$ is a normalized loss at client $c$ evaluated as

$$\mathcal{L}^{(c)}(\theta) = \frac{1}{n_c} \sum_{i=1}^{n_c} \ell \left( h(x_i^c; \theta), x_i^c \right). \tag{3}$$

A min-max problem of Lagrangian associated with a constrained optimization problem can generally be solved using gradient descent-ascent, but the presence of a non-differentiable $L_0$ pseudo-norm poses a challenge for gradient descent-based optimization, a convenient choice for training ML models.

Louizos et al. (2017) present an $L_0$ regularized objective with a set of gates $z \in \mathbb{R}^p$ introduced in a reparameterization of $\theta = \tilde{\theta} \odot z$[1]. Assuming a Hard concrete distribution for sampling gates $z$, the $L_0$ pseudo norm can be approximated as the sum of probabilities of gates being active ($P(z_j = 1)$), enabling the application of the gradient-descent method for the minimization of $L_0$ regularized objective. This method allows passing information about the desired sparsity target through initialization and regularization strength; however, tuning the regularization coefficient to achieve the desired sparsity can be challenging. Gallego-Posada et al. (2022) utilize the same approach, except that they use an $L_0$ density constrained formulation and the min-max problem of the associated Lagrangian. We adopt this approach for our optimization problem in FL due to its flexibility in accommodating generic loss functions and learning controlled sparsity during training through simultaneous gradient descent and ascent.

In this work, we show that the same Lagrangian formulation for the $L_0$ constrained problem in centralized machine learning can be derived from the entropy maximization of the stochastic gates. We then propose a distributed algorithm for learning a sparse global model by aggregating reparameterized gradients across clients. Experiments on synthetic data are used to test the sparsity recovery and statistical performance, as well as the statistical performance at the desired parameter density in real-world datasets. All experiments are conducted on data distributed heterogeneously across clients by design, and with simulation of stagglers or client participation heterogeneity in the training algorithm.

The latter part of the paper is organized into sections on $L_0$- constrained formulation, followed by a distributed algorithm, experiments, and a discussion.

## 2 $L_0$ CONSTRAINED FORMULATION

### 2.1 CENTRALIZED ML

Assuming a linear model $h(x, \theta) : \mathbb{R}^p \to \mathbb{R}$ and loss $\ell(h(x; \theta), y)$ where $x \in \mathbb{R}^p, y \in \mathbb{R}$, and $\theta \in \mathbb{R}^p$, with centralized data $D : (X, Y)$, $X \in \mathbb{R}^{N \times p}$, and $Y \in \mathbb{R}^N$. The $L_0$ density-constrained minimization problem with a desired density of $\rho$ can be described as shown below in a centralized setting.

$$\min_\theta \quad \frac{1}{N} \sum_{i=1}^{N} \ell(h(x_i; \theta), y_i) \quad \text{subject to} \quad \frac{\|\theta\|_0}{|\theta|} \leq \rho, \quad \|\theta\|_0 = \sum_{j=1}^{|\theta|} \mathbb{I}[\theta_j \neq 0] \tag{4}$$

Since the presence of a non-differentiable pseudo-norm makes the application of gradient descent-based methods infeasible on the min-max problem of the Lagrangian, Gallego-Posada et al. (2022)

---

[1] $\odot$ stands for the Hadamard product (Horn, 1990)

use an alternate formulation with reparameterization of $\theta(\tilde{\theta} \odot z)$ using gates $z \in [0,1]^p$ with hard concrete distribution (Louizos et al., 2017) for $z_j$ with parameters $\phi_j$. The minimization objective is an expectation of the loss with respect to the distribution of gates, and the sum of the probabilities of stochastic gates being non-zero is the continuous approximation of the $L_0$ pseudo-norm. $z_j$ being a deterministic transformation of a parameter-free noise allows for a joint optimization of the Monte Carlo approximation of the expected loss over the noise, with respect to $\tilde{\theta}$ and $\phi$ using reparameterized gradients (Ranganath, 2017, ch. 3.3.3). The Lagrangian and the min-max problem associated with the constrained optimization problem with this reparameterization are

$$\hat{\mathfrak{L}}(\tilde{\theta}, \phi; \{\lambda\}) = \sum_{r=1}^{R} \frac{1}{R} \left[ \frac{1}{N} \sum_{i=1}^{N} \mathcal{L} \left( h(x_i; \tilde{\theta} \odot z^{(r)}), y_i \right) \right] + \lambda \left[ \sum_{j=1}^{|\theta|} \frac{E_{q(z|\phi)}[z_j]}{|\theta|} - \rho \right], \text{ and} \quad (5)$$

$$\theta^*, \phi^*, \lambda^* = \arg \min_{\tilde{\theta}, \phi} \arg \max_{\lambda \geq 0} \left( \hat{\mathfrak{L}}(\tilde{\theta}, \phi; \{\lambda\}) \right). \quad (6)$$

## 2.2 ENTROPY MAXIMIZATION OF STOCHASTIC GATES

We use entropy $(H(S) = -\sum_{S \in \Omega} P(S) \log P(S))$ maximization of states $S \in \Omega = \{0,1\}^p$ given a set of constraints to derive the Boltzmann distribution and free energy. Exploiting the connection between free energy upper bound and evidence lower bound (ELBO) (Altosaar et al., 2019), we show that the constrained optimization problem that Gallego-Posada et al. (2022) propose for controlled sparsity naturally arises from the minimization of free energy upper bound with a constraint on the density of micro states. The exploration of such a connection between statistical mechanics and machine learning is not new. LeCun et al. (2006) introduce the energy-based models, where the energy corresponds to the loss of a model. Carbone (2025) review theoretical and practical aspects of energy-based models, connecting elements of statistical physics and machine learning. Lairez (2023) provide a short introduction to the derivation of boltzmann distribution and free energy.

We assume $P(S)$ drives the state of parameters $\theta \in \mathbb{R}^p$ being non-zero with a parameterization of $\theta = \tilde{\theta} \odot S$ $(\tilde{\theta}(\neq 0))$ and thus refer to $S$ as gates. The Lagrangian associated with entropy maximization of $S$, subject to normalization constraint for $P(S)$, expected gate density constraint, and a finite constraint on the normalized loss of a reparameterized model on data $D : (X, Y)$ where $X \in \mathbb{R}^{N \times p}$ and $Y \in \mathbb{R}^N$, can be expressed as

$$\mathfrak{L}(P(S); \{\lambda_i\}) = \sum_{\Omega} P(S) \log P(S) + \lambda_0 \left( \sum_{\Omega} P(S) - 1 \right)$$

$$+ \lambda_1 \left( \sum_{\Omega} P(S) \sum_{j=1}^{|\theta|} \frac{s_j}{|\theta|} - \rho \right) + \lambda_2 \sum_{\Omega} \left( P(S) \left[ \frac{1}{N} \sum_{i=1}^{N} \ell \left( h(x_i; \tilde{\theta} \odot S), y_i \right) \right] - L^* \right). \quad (7)$$

Here, the Lagrange multiplier $\lambda_2$ identified as inverse temperature $\beta = 1/T$ is positive, $\lambda_1$ is non-negative, and $L^*$ is a finite constant. A detailed derivation of the following steps is provided in Appendix A.

The probability distribution $P(S)$ can be expressed in terms of the Hamiltonian $(H(S))$ or energy function in S and the normalizing constant Z using the stationarity condition (Lairez, 2023, eq 17):

$$H(S) = \frac{1}{N} \sum_{i=1}^{N} \ell(h(x_i; \tilde{\theta} \odot S), y_i) + \lambda \sum_{j=1}^{|\theta|} \frac{s_j}{|\theta|}, \quad P(S) = \frac{e^{-\frac{1}{T}H(S)}}{Z}$$

$$Z = \sum_{\Omega} e^{-\frac{1}{T}H(S)} \quad \text{, where } e^{-\frac{1}{T}H(S)} \text{ is the Boltzmann factor.} \quad (8)$$

This distribution is intractable because the normalizing constant is not factorizable in S, resulting in combinatorial complexity. If a simpler $H_0(S)$ such as $\sum_j \lambda s_j h_j / |\theta|$ that factorizes over S is chosen as a trial Hamiltonian, then the associated trial distribution $q(S)$ is a tractable mean field approximation of $P(S)$ and the marginal $q(s_j)$ is a Bernoulli distribution. Using the Bogoliubov variational principle, an upper bound on the free energy can be derived (Kuzemsky, 2015, ch 8). Altosaar et al. (2019) show that minimization of the free energy upper bound is the same as minimization of negative ELBO in the Bayesian variational principle, using the Boltzmann factor as an unnormalized

posterior and the trial distribution as the approximate posterior in ELBO . A lower bound on the negative ELBO ($\mathcal{F}_{LB}$) can further be obtained using the positivity of Kullback-Leibler divergence $\mathcal{D}_{KL}\left(q(S) \mid p(S)\right)$ between the approximate posterior and the prior of the same form. $\mathcal{F}_{LB}$ contains expectations of the normalized loss of the reparameterized model and the gate density, with respect to $q(S)$. The minimization of $\mathcal{F}_{LB}$ involves discrete sampling of the gates from $q(S)$. Since the gradients of the model with respect to parameters of the variational distribution $q(S)$ do not flow through the discrete sampling, a stochastic minimization procedure with Monte-Carlo estimation (Carbone, 2025, eq.27) can be employed, leading to:

$$\hat{\mathcal{F}}_{LB} = \sum_{r=1}^{R} \frac{1}{R} \left[ \frac{1}{N} \sum_{i=1}^{N} \ell \left( h(x_i; \tilde{\theta} \odot S^{(r)}), y_i \right) \right] + \lambda E_{q(S)} \left[ \sum_{j=1}^{|\theta|} \frac{s_j}{|\theta|} \right] \tag{9}$$

The expectation of $\sum_j s_j$ is the sum of the probabilities of non-zero gates, a differentiable relaxation of $L_0$ pseudo-norm counting non-zero parameters given $\tilde{\theta} \neq 0$. Choosing a continuous approximation of the Bernoulli distribution and sampling, that can be expressed as a deterministic transformation of a parameter-free noise such as hard concrete distribution, the Lagrangian for the $L_0$ constrained minimization of $\hat{\mathcal{F}}_{LB}$ is exactly the same as equation 5.

## 2.3 FL WITH REPARAMETERIZATION

In an FL setting with $C$ clients holding data $(D^{(c)})_{c=1}^{C} : (X^c, Y^c)$, a reparameterized linear model $h(x; \tilde{\theta} \odot z) : \mathbb{R}^p \to \mathbb{R}$, and loss $\ell(h(x; \tilde{\theta} \odot z), y)$ where $X^c \in \mathbb{R}^{n_c \times p}$, $Y^c \in \mathbb{R}^{n_c}$, $\sum_{c=1}^{C} n_c = N$, $x \in \mathbb{R}^p$, $y \in \mathbb{R}$, $\theta \in \mathbb{R}^p$ ( $\theta = \tilde{\theta} \odot z$), and gate parameters $\phi = \log \alpha \in \mathbb{R}^p$, the Lagrangian for the $L_0$ density constrained minimization problem can be written as

$$\hat{\mathfrak{L}}(\tilde{\theta}, \phi; \lambda) = \sum_{c=1}^{C} \frac{n_c}{N} \mathcal{L}^{(c)}(\tilde{\theta}, \phi) + \lambda \left( \sum_{j=1}^{|\theta|} \frac{E_{q(z|\phi)}[z_j]}{|\theta|} - \rho \right). \tag{10}$$

Here, $\mathcal{L}^{(c)}(\tilde{\theta}, \phi)$ is Monte Carlo estimate of the normalized loss at client $c$ evaluated as

$$\mathcal{L}^{(c)}(\tilde{\theta}, \phi) = \frac{1}{R} \sum_{r=1}^{R} \frac{1}{n_c} \sum_{i=1}^{n_c} \ell \left( h(x_i^c; \tilde{\theta} \odot z^{(r)}), x_i^c \right). \tag{11}$$

A hard concrete distribution g(f($\phi, \epsilon$)) for sampling of gates($z$)is a hard-sigmoid transformation of the stretched $\bar{s}$ of the binary concrete random variable $s$. Louizos et al. (2017) proposed hard concrete as a closer approximation, allowing for zeros in $z$, of Bernoulli than the binary concrete (Maddison et al., 2016).

$$Concrete: s = q(s|\phi) = \sigma \left( \frac{\log \frac{u}{1-u} + \log \alpha}{\beta'} \right), \quad u \sim \mathcal{U}(0, 1),$$

$$Stretch: \bar{s} = s(\zeta - \gamma) + \gamma, \quad Transform: z = \min(1, \max(0, \bar{s})). \tag{12}$$

Using the cumulative distribution Q($\bar{s}$), presented at Louizos et al. (2017),the min-max problem can be expressed as:

$$\tilde{\theta}^*, \phi^*, \lambda^* = arg \min_{\tilde{\theta}, \phi} arg \max_{\lambda \geq 0} \left( \sum_{c=1}^{C} \frac{n_c}{N} \mathcal{L}^{(c)}(\tilde{\theta}, \phi) + \lambda \left( \sum_{j=1}^{|\theta|} \frac{E_{q(z|\phi)}[z_j]}{|\theta|} - \rho \right) \right) \tag{13}$$

$$\text{where} \quad E_{q(z|\phi)}[z_j] = 1 - Q(\bar{s}_j \leq 0|\phi_j) = \sigma \left( \log \alpha_j - \beta' \log \left( -\frac{\gamma}{\zeta} \right) \right).$$

We can now perform a joint optimization of $\tilde{\theta}$ and $\phi = \log \alpha$ using gradient descent with reparameterized gradients, and a test-time $\hat{z}^r$ without noise and smoothing (Louizos et al., 2017). We use a gradient ascent updating rule for $\lambda$ with restart ($\lambda$=0) when the constraint is satisfied (Gallego-Posada et al., 2022). We also note that a local free energy with constraints, local to the client, can be conceived, where the parameters are optimized at the client level first and aggregated at the server level. The first approach is to minimize the empirical loss at each client that contributes to the global free energy. In contrast, the second approach is akin to reducing local free energies and aggregating parameters. In both cases, we are interested in global free energy minimization.

## 3 DISTRIBUTED OPTIMIZATION ALGORITHM

We now introduce a short notation $\mathcal{L}_{Con}(\phi)$ representing the $L_0$ density constraint in equation 10. The Lagrangian in short notation can be expressed as

$$\hat{\mathfrak{L}}(\tilde{\theta}, \phi; \{\lambda\}) = \sum_{c=1}^{C} \frac{n_c}{N} \mathcal{L}^{(c)}(\tilde{\theta}, \phi) + \lambda \mathcal{L}_{Con}(\phi). \tag{14}$$

McMahan et al. (2017) propose a distributed algorithm for learning a global model with synchronous update using averaging of gradients. Assuming a central server that coordinates training with $C$ clients holding data $(D^c)_{c=1}^{C}$ locally, clients compute the gradients and the server aggregates these gradients from the clients to update the global model. For all data at each client, clients compute $\nabla_{\tilde{\theta}} \mathcal{L}^{(c)}(\tilde{\theta}, \phi)$ and $\nabla_{\phi} \mathcal{L}^{(c)}(\tilde{\theta}, \phi)$ at time $t$ and the server updates: $\tilde{\theta}^{t+1} \leftarrow \tilde{\theta}^t - \eta_{\tilde{\theta}} \sum_{c=1}^{C} \frac{n_c}{N} \nabla_{\tilde{\theta}} \mathcal{L}^{(c)}(\tilde{\theta}^t, \phi^t)$, $\phi^{t+1} \leftarrow \phi^t - \eta_{\phi}(\sum_{c=1}^{C} \frac{n_c}{N} \nabla_{\phi} \mathcal{L}^{(c)}(\tilde{\theta}^t, \phi^t) - \lambda \nabla_{\phi} \mathcal{L}_{Con}(\phi^t))$ and $\lambda^{t+1} \leftarrow \lambda^t + \eta_{\lambda} \mathcal{L}_{Con}(\phi^t)$.

We use mini-batches of uniform size $B$ drawn randomly at all clients for a synchronous update of global parameters, and repeat this process for $n(B)$ iterations in each epoch/round. A fraction $\gamma_c$ of clients C is chosen at random for training in each epoch ($K = \lfloor \gamma_c \cdot C \rfloor$). For a mini-batch at each client, clients compute $\nabla_{\tilde{\theta}} \mathcal{L}_B^{(k)}(\tilde{\theta}, \phi)$ and $\nabla_{\phi} \mathcal{L}_B^{(k)}(\tilde{\theta}, \phi)$, the gradients of mini-batch normalized loss with respect to $\tilde{\theta}$ and $\phi$, and the server updates: $\tilde{\theta}^{t+1} \leftarrow \tilde{\theta}^t - \eta_{\tilde{\theta}} \sum_{k=1}^{K} w_k \nabla_{\tilde{\theta}} \mathcal{L}_B^{(k)}(\tilde{\theta}^t, \phi^t)$, $\phi^{t+1} \leftarrow \phi^t - \eta_{\phi}(\sum_{k=1}^{K} w_k \nabla_{\phi} \mathcal{L}_B^{(k)}(\tilde{\theta}^t, \phi^t) + \lambda \nabla_{\phi} \mathcal{L}_{Con}(\phi^t))$ and same update for $\lambda$ as before but with a restart mechanism if the constraint is satisfied (Gallego-Posada et al., 2022). The weights for aggregation in our case are $w_k = \frac{B}{K \cdot B}$. With this approach, we learn a global sparse model in FL with an $L_0$ constraint using probabilistic gates, which we refer to as FLoPS.

The learning rates $\eta_{\tilde{\theta}} \sim [1e^{-3}, 1e^{-1}]$, $\eta_{\phi} \sim [1e^{-4}, 1e^{-1}]$, and $\eta_{\lambda}$ in the order of $1/|\theta|$ are tuned for stability in training. The gate parameters $\log \alpha$ are sampled from normal distribution with mean of $\log \rho_{\text{init}} - \log(1 - \rho_{\text{init}})$ and variance 0.01 and $\rho_{\text{targ}} = \rho$ is the desired density of parameters. A high $\rho_{\text{init}}$ implies a dense initialization. The hyperparameters of hard concrete: $\gamma$, $\zeta$, and $\beta'$ are set at $-0.1$, 1.1, and 0.66 as recommended (Louizos et al., 2017). The Lagrange parameter $\lambda$ is set to 0 initially.

---

**Algorithm 1** FLoPS. $E$ is the number of epochs and $B$ is the mini-batch size.

---

1: **Server:**
2: Initialize: $(\tilde{\theta}^{(0)}, \phi^{(0)} : \log \alpha^{(0)})$
3: **for** epoch t $= 1, \ldots, E$ **do**
4:     $S_t$=random set of $K$ clients ($K = \lfloor \gamma_c \cdot C \rfloor$)
5:     **for** batch $b = 1, \ldots, n(B)$ **do**
6:         **for** each participating client $k \in S_t$ **do**
7:             gradients with respect to $(\tilde{\theta}, \phi)$: **ClientCompute**$(k, \tilde{\theta}^t, \phi^t)$
8:         **end for**
9:         Aggregate gradients from clients w.r.t $(\tilde{\theta}, \phi)$: $\sum_{k=1}^{K} w_k g_{\tilde{\theta}}^k$ and $\sum_{k=1}^{K} w_k g_{\phi}^k$
10:         Compute gradients $\nabla_{\phi} \mathcal{L}_{Con}(\phi^{(t)})$
11:         Update $(\tilde{\theta}^{(t+1)}, \phi^{(t+1)})$ and the dual $\lambda^{(t+1)}$
12:     **end for**
13:     **If epoch $>$ prune start:**
14:     scale up $\log \alpha^{(b)}$ at top-$m$ indices of $\theta$ ($m = \lfloor (\rho_{\text{targ}} \cdot |\theta|) \rfloor$) and scale down for the remaining.
15:     $((\log \alpha^{(t)} + r \log \alpha^{(t)})m_{\theta} + (\log \alpha^{(t)} - r \log \alpha^{(t)})(1 - m_{\theta})$ where $r$ and $m_{\theta}$
16:     are decay factor and top-$m$ mask respectively.)
17: **end for**
18: **ClientCompute**$(k, \tilde{\theta}, \phi)$: // Run for each client $k \in S_t$
19:     Draw a mini batch $b \leftarrow random(D^{(k)}, B)$ of size $B$
20:     Compute $g_{\tilde{\theta}}^k = \nabla_{\tilde{\theta}} \mathcal{L}_B^{(k)}(\tilde{\theta}^t, \phi^t)$ and $g_{\phi}^k = \nabla_{\phi} \mathcal{L}_B^{(k)}(\tilde{\theta}^t, \phi^t)$
21:     Send $g_{\tilde{\theta}}^k$ and $g_{\phi}^k$ to server

---

---

**Algorithm 2** FLoPS-PA. $E$, $B$ , and $\oslash$ are epochs, mini-batch size and element wise division.

---

1: **Server:** Initialize $(\tilde{\theta}^{(0)}, \phi^{(0)}) : \theta^{(0)} = \tilde{\theta}^{(0)} \odot z^{(0)}$
2: **for** epoch $b = 1 \ldots E$ **do**
3:     **for each** client $k \in S_t$ **do**
4:       $(\theta_k, z_k)$:**ClientCompute**$(k, \theta^{(t)}, z^{(t)})$
5:     **end for**
6:     **Server aggregation (parameter averaging):**
7:       $\theta = \sum_{k \in S_t} w_k \theta_k$
8:       $z = \sum_{k \in S_t} w_k z_k$
9:       $\tilde{\theta} = \theta \oslash z$ and get $\phi$ from $z$
10:     Server side tuning of $\tilde{\theta}, \phi$
11:     **if** epoch $>$ prune start **then**
12:       scale up $\phi : \log \alpha$ at top-$m$ indices of $\theta$ ($m = \lfloor (\rho_{\text{targ}} \cdot |\theta|) \rfloor$) and scale down for the
13:     remaining.
14:     **end if**
15: **end for**

16: **ClientCompute**$(k, \theta, z)$: // Run for each client $k \in S_t$
17:     $\tilde{\theta}_k = \theta \oslash z$ and $\phi_k$ is obtained from $z$
18:     Perform n(B) local SGD steps on $(\tilde{\theta}_k, \phi_k)$ drawing $b \leftarrow random(D^{(k)}, B)$ of size $B$
19:     Sample gates $z_k$ and compute $\theta_k$
20:     Return $(\theta_k, z_k)$

---

By design, FLoPS gives test time sparsity with $\hat{z}$ sampled without noise and smoothing. Gallego-Posada et al. (2022) note that their approach results in a density closer to the targeted density. For exact sparsity, while not breaking the assumption of $\tilde{\theta} \neq 0$, we scale up $\log \alpha$ at top-$m$ indices where $m = \lfloor \rho_{\text{targ}} \cdot |\theta| \rfloor$ and scale down for the rest of the index in each epoch after a set threshold called prune start. The fact that not all clients may qualify to participate in training is simulated by randomly choosing a fraction of clients for training in each epoch. In an FL setting, since the data available at a client varies, application of methods such as the Reimannian aggregation scheme (Ahmad et al., 2023) or simply choosing $w_k = \frac{n_k}{N}$, can be employed. Zhao et al. (2018) note that the data heterogeneity in FL leads to weight divergence, resulting in reduced statistical performance, and propose utilizing a small portion of data at the server to tune the aggregated model. The algorithm FLoPS involves communicating full gradient vectors from clients to the server at every epoch and each step before updating the global model with gradient averages. This is communication-heavy, in message size and number of times of communication and thus we propose a federated averaging style FLoPS-PA which reduces communication rounds to once per epoch. We also further compress the message size, reducing uplink and downlink communication. In this regard, we start scaling $log\alpha$ at the server and the client before communication begins from epoch zero. The parameters $\theta$ beyond top$-m$ indices are set to zero and $z$ (sampled using $\log \alpha$) beyond top$-m$ indices are replaced with their average ($z_{-m}^{avg}$). The top-$m$ $\theta$ and $z$ with their indices, along with one additional value of $z_{-m}^{avg}$, are communicated. At the receiving end $\tilde{\theta} = \theta \oslash z$ and $\log \alpha = \beta' \log(z/1 - z)$ are computed ignoring the noise component. The communication cost reduction in this way is significant, especially for small $\rho_{targ}$.

## 4 EXPERIMENTS

The experiments conducted span both synthetic and real datasets. The sparse linear regression (LR), logistic regression (LG), and multi-class classification (MC) on synthetically generated data are included. The experimental results of MC, and multi-label classification (MLC) on real datasets MNIST, EMNIST, and RCV1 are also included. We conducted all our experiments on an Apple MacBook with an M4 Pro chip (12-core CPU) and 24 GB of unified memory running macOS 15.5, using PyTorch (2.7.0) in Python (3.12.7).

| Model | Density ($\rho_{\text{true}}$) | TDR | | | Epoch 50: Statistical performance (test time) | | | | | |
|---|---|---|---|---|---|---|---|---|---|---|
| | | FLoPS(0.05) | FLoPS(0.95) | FedIter-HT | FLoPS(0.05) | | FLoPS(0.95) | | FedIter-HT | |
| | | | | | $R^2$/ACC | MSE/CE | $R^2$/ACC | MSE/CE | $R^2$/ACC | MSE/CE |
| LR | 0.05 | 1.00 | 0.05 | 0.18 | 0.91 | 4.62 | 0.85 | 7.65 | 0.16 | 42.57 |
| LG | 0.05 | 0.94 | 0.11 | 0.11 | 0.90 | 0.32 | 0.83 | 0.56 | 0.63 | 0.88 |
| MC | 0.05 | 0.99 | 0.38 | 0.51 | 0.68 | 0.82 | 0.52 | 2.25 | 0.24 | 2.24 |

Table 1: This table corresponds to synthetic data generated using a signal-to-noise (SNR) ratio of 20 and a covariance matrix generated using a 0.2 correlation factor with a true model density of 0.05. It provides a comparison across models learned under data and client participation heterogeneity. On the left: TDR for `FLoPS` trained to achieve densities 0.05 and 0.95, and `FedIter-HT` trained to achieve true density; On the right: Statistical performance at test time for all models showing $R^2$ and MSE for LR, and Accuracy (ACC) and CE loss for classification models after training for 50 epochs.

## 4.1 EXPERIMENTS ON SYNTHETIC DATA SETS

For generating synthetic linear regression data, we use a sparse linear model following the method described at Bertsimas et al. (2020). Each row $x_i \in \mathbb{R}^{1000}$ of the design matrix $X \in \mathbb{R}^{10000 \times 1000}$ is drawn from a zero-mean Gaussian with a covariance matrix $\Sigma$. We use a Toeplitz covariance matrix $\Sigma$ where $(\Sigma_{ij})_{i,j=1}^{1000} = \rho_{cor}^{|i-j|}$. An $m$-sparse ($m = \lfloor \rho \cdot 1000 \rfloor$) coefficient vector $w_{\text{true}} \in \mathbb{R}^{1000}$ is constructed by randomly choosing a set of $m$ indices ($S_m \subseteq \{1, ..1000\}$ and sampling $(w_{\text{true}})_j \sim \text{Unif}\{-1, 1\}$ for $j \in S_m$ and $(w_{\text{true}})_j = 0$ otherwise. This is the true sparsity of the model. The responses or predictions are generated as $y = Xw_{\text{true}} + \varepsilon$ with i.i.d. noise $\varepsilon \sim \mathcal{N}(0, \sigma^2 I_N)$ where $N = 10000$. A signal-to-noise ratio defined by $\text{SNR} = \|Xw_{\text{true}}\|_2^2 / \|\varepsilon\|_2^2$ is choosen and the noise level $\sigma = \|Xw_{\text{true}}\|_2 / (\sqrt{\text{SNR}} \sqrt{N})$ is chosen accordingly. For generating synthetic sparse logistic regression, the same procedure is used except the binary labels are obtained by thresholding noisy logits ($n_l = Xw_{\text{true}} + \varepsilon$) by 0, i.e., $y_i = \mathbf{1}$ if $(n_l)_i > 0$ and zero otherwise. In the case of multi-class classification ($n_c$ classes), an $m$-sparse ($m = \lfloor \rho \cdot 1000 \cdot n_c \rfloor$) coefficient matrix is constructed by randomly choosing $m$ positions in the matrix to populate using samples from $\text{Unif}\{-1, 1\}$ and zero otherwise. The labels are generated by taking the index of the largest among the noisy logits across classes ($y_i = \arg\max_c\{((n_l)_i)_c\}$).

We employ affine shifting and the Dirichlet partition protocol for simulating data heterogeneity, and randomly sample a fraction of clients in each epoch for client participation heterogeneity in our experiments (Solans et al., 2024; Reisizadeh et al., 2020). We refer readers to Appendix A for more details. The True Discovery Rate (TDR) (Bertsimas et al., 2020) is used as a metric, along with mean squared error (MSE) and $R^2$ for linear regression, and cross-entropy (CE) loss and accuracy for classification, to compare the accuracy of sparsity recovery and statistical performance. We compare our results with the federated iterative hard thresholding algorithm (`FedIter-HT`) proposed by Tong et al. (2022), where a hard thresholding operator external to federated averaging with gradient descent is used to iteratively impose top-$m$ magnitude selection to minimize $L_0$ regularized objective, showing promising results in their experiments. We tested the sparsity recovery and performance of all algorithms with varying correlation factor $\rho_{\text{cor}}$ and SNR in synthetic data, ranging from high to low. The results for the case with low $\rho_{\text{cor}}$ and high SNR for LR, LG, and MC at true sparsity of 5% are illustrated in Table 1. In all three cases, `FLoPS` performs better than `FedIter-HT`, and 5% dense `FLoPS` performs better than 95% dense `FLoPS`. Since we observed similar comparative results, the sparsity recovery results in other conditions are attached in Appendix A to avoid redundancy.

We conducted experiments in homogeneous (IID) and heterogeneous (Non-IID) data distribution with a 10% of client participation in both settings, controlled by the Dirichlet parameter $\alpha_{\text{iid}}$, at different levels of true sparsity, specifically 5% : $\rho_{\text{targ}} = 0.95$ and 95% : $\rho_{\text{targ}} = 0.05$. Here $\rho_{\text{targ}}$ is the desired density of the model. The Table 2 illustrates that `FLoPS` has superior statistical performance ($R^2$/Accuracy(ACC)), and the sparsity recovery accuracy (TDR) consistently, especially at learning models with very low density of parameters or high sparsity.

The desired property of gradually achieving the target density of parameters during training time can be observed through a reduction in the expected number of gates, which is a continuous approxima-

| Model | Density ($\rho_{\text{targ}} = \rho_{\text{targ}}$) | $\alpha_{\text{iid}} = 1000$ (IID) | | | | $\alpha_{\text{iid}} = 0.5$ (non-IID) | | | |
| --- | --- | --- | --- | --- | --- | --- | --- | --- | --- |
| | | FLoPS | | FedIterHT | | FLoPS | | FedIterHT | |
| | | $R^2$/ACC | TDR | $R^2$/ACC | TDR | $R^2$/ACC | TDR | $R^2$/ACC | TDR |
| LR | 0.95 | 0.83 | 0.98 | 0.86 | 0.97 | 0.69 | 0.96 | 0.74 | 0.96 |
| | 0.05 | 0.90 | 1.00 | 0.37 | 0.37 | 0.91 | 1.00 | 0.27 | 0.18 |
| LG | 0.95 | 0.87 | 0.96 | 0.87 | 0.96 | 0.85 | 0.95 | 0.81 | 0.96 |
| | 0.05 | 0.89 | 0.96 | 0.70 | 0.22 | 0.90 | 0.94 | 0.65 | 0.11 |
| MC | 0.95 | 0.52 | 1.00 | 0.53 | 1.00 | 0.50 | 1 | 0.50 | 1.00 |
| | 0.05 | 0.71 | 0.99 | 0.28 | 0.51 | 0.68 | 0.99 | 0.24 | 0.5 |

Table 2: Comparison of FLoPS and FedIter-HT across models and densities (true model sparsity). Here, the sub-columns $R^2$/ACC and TDR(true discovery rate) showcase the statistical performance and accuracy in sparsity recovery, with client participation heterogeneity: $10\%$ of clients participate in training.

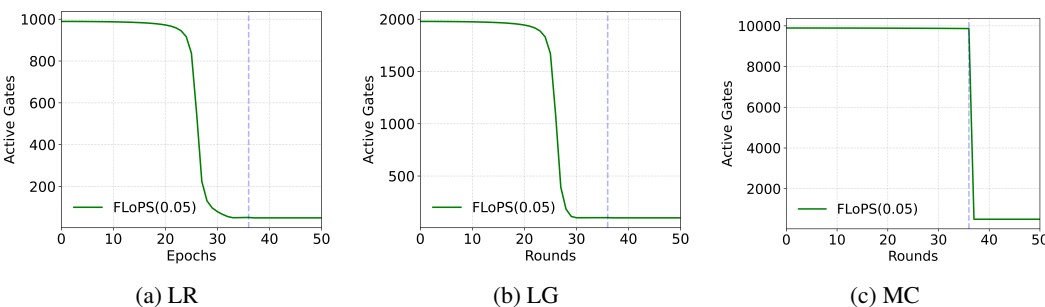

(a) LR          (b) LG          (c) MC

Figure 1: The figure corresponds to results for synthetic data generated using a signal-to-noise (SNR) ratio of 20 and a covariance matrix generated using a 0.2 correlation factor. Here, (a) to (c) correspond to the expected gates of FLoPS: achieving $5\%$ target density of gates during training in heterogeneous conditions of data and client participation (HTC) in LR, LG, and MC cases, respectively. The blue dotted line corresponds to the round at which $\log \alpha$ scaling starts, corresponding to the target density.

tion of the $L_0$ pseudo-norm in FLoPS, over the training epochs. Figure 1 illustrates the reduction in the expected number of gates at test time (active gates) towards the desired sparsity, demonstrating controlled sparsity learning for a target density of $0.05$.

The dynamic sparsity learning in FLoPS can be understood from the change in the sparsity pattern through soft Jaccard loss/ soft intersection over union (IOU) heat map (Wang et al., 2024) of test time gates across epochs, in contrast to the IOU heat map of a binary mask in FedIter-HT where a lower value indicates higher mobility in learning sparsity. The Figure 2 shows a stable, low learning phase in the beginning, starting from a dense initialization, followed by an active learning phase and a stable sparsity pattern towards the end in FLoPS as opposed to continuous change in IOU in FedIter-HT. The heat maps correspond to learning a 0.05 target density of parameters.

The federated averaging algorithm FLoPS-PA allows for sparse communication throughout the training with out any loss of statistical performance. The Figure 4 shows FLoPS-PA achieves better test accuracy compared to FedIter-HT, equivalent performance to FedAvg(dense training and magnitude pruning in the last epoch). An upper bound on test accuracy of sparse models is also presented using Central FLoPS-PA. An estimate of uplink/downlink communication cost can be obtained by multiplying the message size by the number of communication rounds assuming 4 bytes per parameter. It is highest for FedAvg: epochs $\times 4|\theta|$, followed by FLoPS-PA: epochs $\times 4 \times (2 \times \rho_{targ}|\theta|)$ and FedIter-HT: epochs $\times 4 \times \rho_{targ}|\theta|$. The communication cost of FLoPS-PA is twice that of FedIter-HT but significantly lower than dense training. For a large model size $|\theta|$ and low target density $\rho_{targ}$ the cost difference between FedAvg and FLoPS-PA is significantly higher and marginally higher than FedIter-HT. With regards to learning rates for FLoPS-PA , $\eta_\lambda$ is fixed at $0.1/|\theta|$, $\eta_{\tilde\theta} \sim [1e^{-4}, 1e^{-1}]$, and $\eta_\phi \sim [1e^{-5}, 1e^{-1}]$ for stable learning dynamics. The

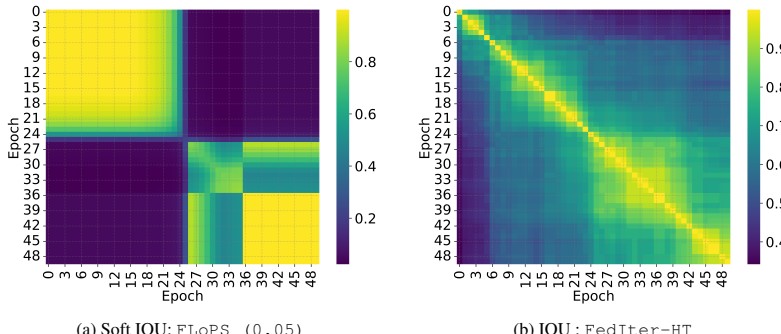

(a) Soft IOU: `FLoPS (0.05)`          (b) IOU: `FedIter-HT`

Figure 2: The figure corresponds to results for synthetic data generated using a signal-to-noise (SNR) ratio of 20 and a covariance matrix generated using a 0.2 correlation factor. Here, (a) and (b) correspond to the soft IOU heat map for test time gates in `FLoPS` for $5\%$ target density of gates and IOU heat map of binary masks in `FedIter-HT` for the same level target density during training in heterogeneous conditions of data and client participation (HTC) in the LR case, respectively.

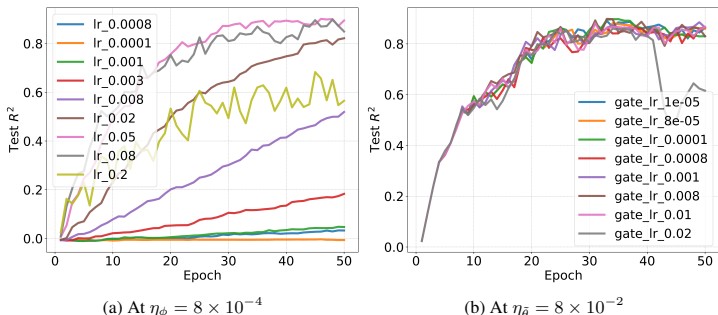

(a) At $\eta_\phi = 8 \times 10^{-4}$          (b) At $\eta_{\tilde{\theta}} = 8 \times 10^{-2}$

Figure 3: The figure corresponds to test $R^2$ of `FLoPS-PA` for various weights $\eta_{\tilde{\theta}}$ and gates $\eta_\phi$ learning rates while keeping the other constant for LR with $5\%$ target density on synthetic data under data and client participation heterogeneity.

Figure 3 shows a wide range of stable gate and parameter learning rates in LR case. The Figure 4 shows empirically that `FLoPS-PA` converges faster than `FedIter-HT` in both heterogeneous and homogeneous data conditions. While the convergence of `FedIter-HT` slows under heterogeneity, `FLoPS-PA` remains with the same rate of convergence. These results are produced from a multi-run experiment with 50 random instances of LR data distributed heterogeneously (homogeneously) across clients, with a client participation rate of $5\%$.

## 4.2 EXPERIMENTS ON REAL DATASETS

We considered three publicly available datasets: *(i)* RCV1 (Lewis et al., 2004) is a multi-label classification dataset with a tfidf representation of Reuters newswire articles as features; we used 34 labels that account for $\sim 87\%$ of all label assignments for the multi-label classification experiment. *(ii)* MNIST (LeCun, 1998) is a multi-class classification dataset of handwritten digits with grayscale image pixel values as features. *(iii)* EMNIST (Cohen et al., 2017) is an extended MNIST data set with upper and lower case letters in addition to digits, with 62 classes in total.

For conducting FL experiments, the data needs to be decentralized. Tong et al. (2022) use k-means clustering to group the data into 10 clusters and partition each cluster into 20 even parts for RCV1 data. A random selection of two clusters is used to sample one partition from each cluster to allocate to one of the 100 clients. We distribute two randomly selected cluster-partition pairs to each of the 100 clients. In this way, each client sees at most two clusters simulating heterogeneity. For MNIST and EMNIST data, the Dirichlet partition protocol (DPP) is applied to generate heterogeneous label proportions for each client. The data is distributed according to DPP across 100 and 1000 clients

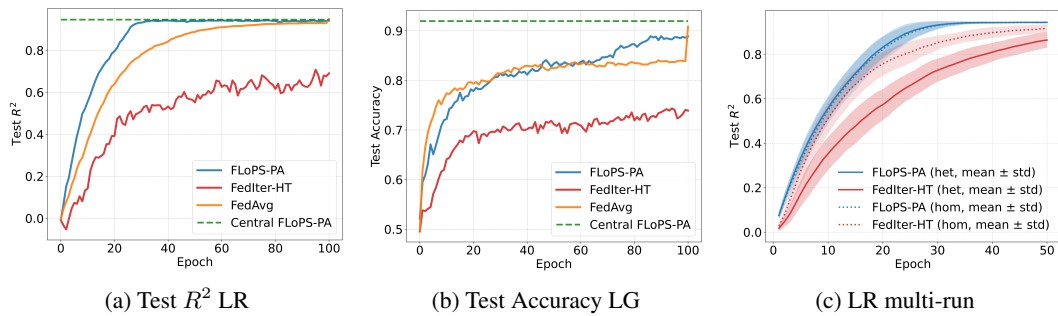

| (a) Test $R^2$ LR | (b) Test Accuracy LG | (c) LR multi-run |

Figure 4: The figure corresponds to results of `FLoPS-PA` and `FedIter-HT` with $5\%$ target density on synthetic LR and LG data, along with test $R^2$ in multi-run experiments with LR.

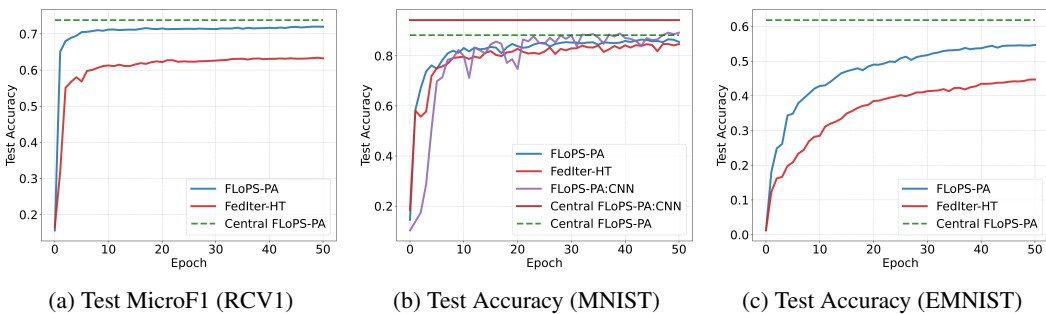

| (a) Test MicroF1 (RCV1) | (b) Test Accuracy (MNIST) | (c) Test Accuracy (EMNIST) |

Figure 5: The figure corresponds to results of `FLoPS-PA`, `FedIter-HT`, and `Central FLoPS-PA` with on RCV1 at target density of $0.005\%$, MNIST and EMNIST data at target density of $5\%$ at heterogeneous conditions of data and client participation.

for MNIST and EMNIST, respectively. (Solans et al., 2024). In each epoch, only $5\%$ of clients are randomly sampled to simulate heterogeneity in client participation.

With $47236$ dimensions, the RCV1 has high-dimensional and sparse features, leading to an MLC model with a size of $\sim 1.6$ million parameters. The MNIST data has $28 \times 28$ pixel values as features, leading to an MC model size of $7840$ parameters. With a CNN with two convolution layers(6 channels - 16 channels with kernel size 5) with max $2 \times 2$ max pooling layers and three fully connected layers (256,120,84) leading to a model size of $44426$. The EMNIST data has the same input dimensions, leading to an MC with $48608$ parameter size. The binary CE (BCE)-micro-averaged $F_1$, and CE-accuracy are used for comparing the statistical performance of MLC, and MC on MNIST and EMNIST, respectively. Figure 5 shows that `FLoPS-PA` has a superior test time performance compared to `FedIter-HT` in the RCV1 (MLC:$\rho_{targ} = 0.5\%$), MNIST (MC and CNN MC:$\rho_{targ} = 5\%$), and EMNIST (MC: $rho_{targ} = 0.5\%$) cases with higher micro-averaged F1 and multi-class classification accuracies.

## 5 CONCLUSION

We introduced $L_0$ density-constrained based optimization in FL for learning a global model with desired sparsity using a reparameterization with probabilistic gates. We showed that the $L_0$ regularized objective (Louizos et al., 2017) and the $L_0$ constrained formulation (Gallego-Posada et al., 2022) can be derived from entropy maximization of stochastic gates introduced for inducing sparsity. We reformulated the min-max problem associated with the Lagrangian for $L_0$ constrained optimization for FL. We proposed the distributed optimization algorithms `FLoPS` and `FLoPS-PA` and evaluated them using synthetic and high-dimensional real data. we showed that `FLoPS-PA` achieves superior statistical performance under data and client participation heterogeneity with low communication cost. Our approach currently assumes a centrally orchestrating server and we intend to adapt it to other types of connectivity.

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

# A APPENDIX

DERIVATION OF OPTIMAL DISTRIBUTION $P(S)$

We start with the Lagrangian for maximizing entropy of the states $S$ or gates with a normalizing constraint, constraint on expectation of density of states, and a constraint on the loss $\ell(h(x; \tilde{\theta} \odot \phi), y)$ of the reparameterized model $(h(x; \tilde{\theta} \odot S) : \mathbb{R}^p \to \mathbb{R})$ normalized over data $D : (X, Y)$ where $x \in \mathbb{R}^p$, $y \in \mathbb{R}$, $X \in \mathbb{R}^{N \times p}$, $Y \in \mathbb{R}^N$ and $\theta \in \mathbb{R}^p (\theta = \tilde{\theta} \odot S)$. The Lagrange multiplier $\lambda_2$, identified as inverse temperature $\beta = 1/\text{T}$, is positive, $\lambda_1$ is non-negative, and $L^*$ is a finite constant.

$$
\mathfrak{L}(P(S); \{\lambda_i\}) = \sum_{\Omega} P(S) \log P(S) + \lambda_0 \left( \sum_{\Omega} P(S) - 1 \right)
$$

$$
+ \lambda_1 \left( \sum_{\Omega} P(S) \sum_{j=1}^{|\theta|} \frac{s_j}{|\theta|} - \rho \right) + \lambda_2 \sum_{\Omega} \left( P(S) \left[ \frac{1}{N} \sum_{i=1}^{N} \ell \left( h(x_i; \tilde{\theta} \odot S), y_i \right) \right] - L^* \right). \quad (15)
$$

Taking the functional derivative w.r.t. $P(S)$ to zero, a stationarity condition, and solving for $p(S)$ gives the Gibbs-Boltzmann distribution.

$$\frac{\partial \mathfrak{L}(P(S); \{\lambda_i\})}{\partial P(S)} = \frac{1}{N} \sum_{i=1}^{N} \ell \left( h(x_i; \tilde{\theta} \odot S), y_i \right) + T(1 + \log P(S)) + \lambda \sum_{j=1}^{|\theta|} \frac{s_j}{|\theta|} + \mu = 0.$$

Here, $\mu = \lambda_0 T$ and $\lambda = \lambda_1 T$. Solving for $\log P(S)$:

$$\log P(S) = -\frac{1}{T} \left[ \frac{1}{N} \sum_{i=1}^{N} \ell \left( h(x_i; \tilde{\theta} \odot S), y_i \right) + \lambda \sum_{j=1}^{|\theta|} \frac{s_j}{|\theta|} + \mu + T \right]$$

$$P(S) = \frac{1}{Z} \exp \left( -\frac{1}{T} \left[ \frac{1}{N} \sum_{i=1}^{N} \ell \left( h(x_i; \tilde{\theta} \odot S), y_i \right) + \lambda \sum_{j=1}^{|\theta|} \frac{s_j}{|\theta|} \right] \right)$$

$$= \frac{1}{Z} \exp \left( -\frac{1}{T} H(S) \right). \tag{16}$$

Equation 16 is the Gibbs-Boltzmann distribution over states for known parameters and data. Here $Z$ is the normalization constant and $H(S)$ is the Hamiltonian or the energy function given by equation 17 and equation 18:

$$Z = \sum_{\Omega} \exp \left( -\frac{1}{T} \left[ \frac{1}{N} \sum_{i=1}^{N} \ell \left( h(x_i; \tilde{\theta} \odot S), y_i \right) + \lambda \sum_{j=1}^{|\theta|} \frac{s_j}{|\theta|} \right] \right). \tag{17}$$

$$H(S) = \frac{1}{N} \sum_{i=1}^{N} \ell(h(x_i; \tilde{\theta} \odot S), y_i) + \lambda \sum_{j=1}^{|\theta|} \frac{s_j}{|\theta|} \tag{18}$$

GIBBS-BOGOLIUBOV INEQUALITY AND ELBO

The Helmholtz free energy functional F is defined using logarithmic transformation of the partition function Z and inverse temperature $\beta$. The minimization of free energy yields the probability distribution with maximum entropy.

$$F = -\frac{1}{\beta} \log(Z).$$

The partition function Z associated with $P(S)$ in equation 17 is intractable. A distribution q(S) with a partition function fully factorizable in states $s_i \; \forall i \in \{1, \ldots |\theta|\}$ can be assumed as a trial distribution.

$$q(S) = \frac{1}{Z_0} \exp \left( -\frac{1}{T} H_0(S) \right); \quad F_0 = -\frac{1}{\beta} \log(Z_0)$$

Using the Gibbs-Bogoliubov inequality

$$F \leq F_0 + \mathbb{E}_{Q(S)}[H(S) - H_0(S)] \tag{19}$$

and the relation between free energy and the entropy

$$F_0 = \mathbb{E}_{q(S)}[H_0(S)] - T\mathcal{H}[q], \tag{20}$$

an upper bound on true free energy can be obtained (Kuzemsky, 2015, eq 22, eq 122). This is the variational free energy upper bound expressed in

$$F \leq \mathbb{E}_{q(S)}[H(S)] - T\mathcal{H}[q]. \tag{21}$$

Altosaar et al. (2019) show the relation between the Gibbs-Bogoliubov inequality for a system with null data using a mean field (MF) trial distribution and the evidence lower bound (ELBO), where

the unnormalized posterior distribution corresponds to the Boltzmann factor ($e^{-\frac{1}{T}E}$) in Gibbs-Boltzmann distribution with energy function E. The ELBO can be expressed using the log joint or the unnormalized posterior corresponding to the Gibbs-Boltzmann factor in our context:

$$
\begin{aligned}
\mathbb{L} &= \mathbb{E}_{q(S)}[\log P(D, S)] - \mathbb{E}_{q(S)}[\log q(S)] \\
&= \mathbb{E}_{q(S)}[\log P(D|S)] - \mathbb{E}_{q(S)}[\log q(S)] + \mathbb{E}_{q(S)}[\log p(S)] \\
&= \mathbb{E}_{q(S)}[\log P(D|S)] - \mathcal{D}_{KL}[q(S)|p(S)].
\end{aligned}
$$

The minimization of negative ELBO or variational free energy is the same as the minimization of the upper bound on the free energy described in equation equation 21 (Altosaar et al., 2019):

$$
\begin{aligned}
F = -\mathbb{L} &= \mathbb{E}_{q(S)}[-\log P(D|S)] + \mathcal{D}_{KL}[q(S) \mid p(S)] \\
&\geq \mathbb{E}_{q(S)}[-\log P(D|S)] = \mathcal{F}_{LB}
\end{aligned}
\tag{22}
$$

MINIMIZATION OF THE BAYESIAN VARIATIONAL FREE ENERGY

The minimization of the lower bound $\mathcal{F}_{LB}$ on the Bayesian variational free energy derived using the positivity of $\mathcal{D}_{KL}(q(S) \mid p(S))$ involves sampling the state variables from q(S). For a choice of $H_0(S) = \sum_j \lambda s_j h_j / |\theta|$, q(S) is a mean field approximation of $P(S)$ and fully factorizable in S. The probability distribution $q(s_j)$ is a Bernoulli probability. Since the gradients do not flow through the discrete sampling from the Bernoulli distribution, a stochastic minimization procedure with Monte-Carlo estimation, although inefficient, can be employed (Carbone, 2025, eq.27):

$$
\begin{aligned}
\hat{\mathcal{F}}_{LB} &= \sum_{r=1}^{R} \frac{1}{R} \big[ -\log P(D \mid S^{(r)}) \big] \\
&= \sum_{r=1}^{R} \frac{1}{R} \left[ \frac{1}{N} \sum_{i=1}^{N} \ell \left( h(x_i; \tilde{\theta} \odot S^{(r)}), y_i \right) \right] + \lambda E_{q(S)} \left[ \sum_{j=1}^{|\theta|} \frac{s_j}{|\theta|} \right]
\end{aligned}
\tag{23}
$$

Ranganath (2017) provide a recipe for efficiently working with stochastic optimization by employing variance reduction methods such as reparametrized gradients. However, this approach excludes the usage of discrete latent random variables S and assumes access to the gradient of the log joint or the model with respect to latent variables. Further assuming that the sampling of continuous latent variables S or gates can be expressed as a deterministic transformation of a parameter-free noise $\epsilon$, one can simplify the stochastic optimization at hand as joint optimization of the model parameters and the gate parameters using reparameterized gradients:

$$
S^{(r)} = f(\phi, \epsilon^{(r)})
$$
$$
\epsilon^{(r)} \sim p(\epsilon).
$$

Louizos et al. (2017) propose a hard concrete distribution(g(f($\phi$, $\epsilon$))) for continuous sampling of $z$, allowing for a closer approximation of the Bernoulli distribution:

$$
\hat{\mathcal{F}}_{LB} = \sum_{r=1}^{R} \frac{1}{R} \left[ \frac{1}{N} \sum_{i=1}^{N} \ell \left( h(x_i; \tilde{\theta} \odot z^{(r)}), y_i \right) \right] + \lambda E_{q(z|\phi)} \left[ \sum_{j=1}^{|\theta|} \frac{z_j}{|\theta|} \right].
\tag{24}
$$

Here, $z$ is a hard-sigmoid transformation of the stretched $\bar{s}$ of the binary concrete random variable $s$. Using the cumulative distribution $Q(\bar{s})$ this expression can be expressed as (Louizos et al., 2017):

$$
\hat{\mathcal{F}}_{LB} = \sum_{r=1}^{R} \frac{1}{R} \left[ \frac{1}{N} \sum_{i=1}^{N} \ell \left( h(x_i; \tilde{\theta} \odot z^{(r)}), y_i \right) \right] + \lambda \left[ \sum_{j=1}^{|\theta|} \frac{\sigma \left( \log \alpha_j - \beta' \log \left( -\frac{\gamma}{\zeta} \right) \right)}{|\theta|} \right],
\tag{25}
$$

where, $z^{(r)} = \min(1, \max(0, \bar{s}^{(r)}))$, $\bar{s}^{(r)} = s^{(r)}(\zeta - \gamma) + \gamma$, $s^{(r)} = q(s^{(r)} \mid \phi) = \sigma \left( \log \alpha^{(r)} + \log \left( \frac{u^{(r)}}{1-u^{(r)}} \right) \right)$, $u^{(r)} \sim \mathcal{U}(0, 1)$, and $E_{q(z|\phi)}[z_j] = 1 - Q(\bar{s}_j \leq 0) = \sigma \left( \log \alpha_j - \beta' \log \left( -\frac{\gamma}{\zeta} \right) \right)$.

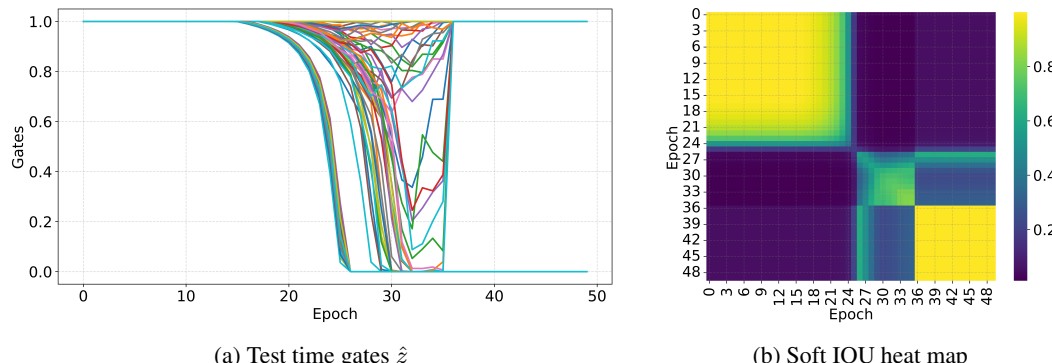

(a) Test time gates $\hat{z}$          (b) Soft IOU heat map

Figure 6: The figure corresponds to `FLoPS(0.05)` for synthetic linear regression data generated using a signal-to-noise (SNR) ratio of 20 and a Covariance matrix generated using a 0.2 correlation factor $\rho_{targ} = 0.05$. Here, (a) and (b) correspond to the test time gates $\hat{z}$ and Soft IOU (Intersection Over Union) heat map of $\hat{z}$ over epochs under data (non-IID) and client participation heterogeneity (10%).

LEARNING SPARSITY THROUGH TEST TIME GATES $\hat{z}$

Figure 6 illustrates how `FLoPS` learns the desired sparsity, using test time gates over epochs and soft jaccard loss/IOU (Wang et al., 2024). The lower the soft IOU, the higher the change or learning in the sparsity pattern.

MORE EXPERIMENTS ON SYNTHETIC DATA

Figure A shows the sparsity recovery accuracy (TDR) of `FLoPS` and `FedIter-HT` in additional experimental conditions to those shown in the experiments section. We generated synthetic data by varying the SNR and the Toeplitz covariance matrix ($\Sigma_{ij} = \rho_{\text{cor}}^{|i-j|}$) for various values of correlation factor $\rho_{\text{cor}}$. We generated results in four regimes, varying SNR from 20 (high) to 3 (low) and $\rho_{\text{cor}}$ from 0.2 (low) to 0.7 (high): *(i)* high SNR-low $\rho_{\text{cor}}$ *(ii)* low SNR-low $\rho_{\text{cor}}$ *(iii)*low SNR-high $\rho_{\text{cor}}$ *(iv)* high SNR-high $\rho_{\text{cor}}$. The results for the first regime are discussed in the experiments section, and the results for the remaining regimes are presented in Figure A. We see that the sparsity recovery is consistently better with `FLoPS` in experiments for the LR, LG, and MC tasks. We also note that the `FLoPS` with 95% target density is trivially poor in sparsity recovery, as we forced the higher density of parameters despite our knowledge of the inherent sparsity. Cherepanova et al. (2023) show that the statistical performance drops with an increase in corrupted or duplicated features. We suspect that the significantly poor performance of `FedIter-HT` in our synthetic data with dense correlated features, in contrast to real data with sparse features, could stem from the same reason and needs further investigation.

HETEROGENEITY

In FL, the data is neither centralized nor independent and identically distributed (IID). The data that each client holds is private and may not have the same quantity or attributes of data as the other clients. In practice, the same number of clients may not be eligible or available to participate throughout the training time. Solans et al. (2024) reviews various reasons for the above-mentioned forms of heterogeneity (non-IIDness) and different ways to simulate such conditions. To simulate heterogeneity in the number of samples across clients or quantity skew, a Dirichlet distribution with parameter $\alpha$ is used to sample proportions for each client. The samples are then allocated to clients according to the sampled proportions, using a Dirichlet partition protocol to achieve quantity skew. For a high $\alpha_{\text{iid}} = 1000$, the samples are uniformly distributed across 100 clients, whereas a low $\alpha_{\text{iid}} = 0.5$ results in a heterogeneous distribution of samples. For achieving attribute skew, affine shifts that are randomly sampled using a zero-mean Gaussian distribution with a standard deviation $\sigma_{ms}$ are then used to shift features at a client by samples from a Gaussian distribution with an affine shift as mean and standard deviation of 1. Reisizadeh et al. (2020) discuss the decline in performance

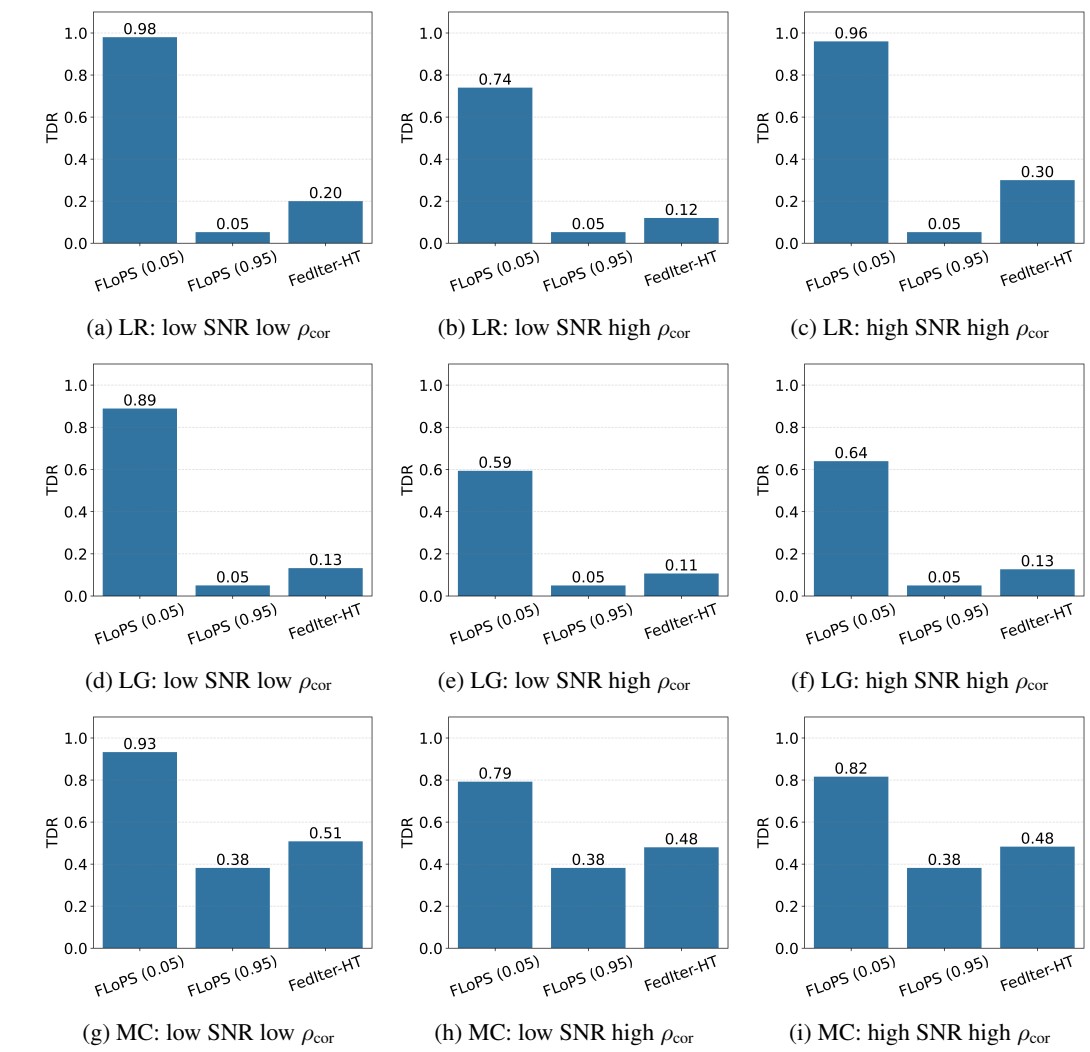

Figure 7: The figure corresponds to sparsity recovery results of `FLoPS` and `FedIter-HT` in data and client participation heterogeneous conditions for synthetic linear regression (LR), logistic (LG) and multiclass classification (MC) data generated in low SNR (3) - low $\rho_{cor}$ (0.2), low SNR (3) - high $\rho_{cor}$ (0.7), and, high SNR (20) - high $\rho_{cor}$ (0.7).

of the model in FL settings with attribute skew simulated using affine shifts. Finally, the client participation skew is simulated by randomly sampling 5% of the 100 clients in each round/epoch.

### CONCRETE DISTRIBUTION

Maddison et al. (2016) propose the binary concrete distribution with parameters $\log \alpha_j$, using a Gumbel max trick on the Bernoulli distribution. Huijben et al. (2022) present a review of the Gumbel max trick in machine learning as a method to generate continuous samples from a deterministic transformation of an IID noise that results in the categorical probabilities. By definition, $\log \alpha_j$ are the logits of the Bernoulli probabilities $\pi_j$ and can be initialized using a normal distribution with a mean of $\log(\rho/1-\rho)$ where $\rho$ determines the expected number of non-zero parameters through the reparameterization, implying dense to sparse and sparse to sparse model training is possible.

$$\log \alpha_j^{(0)} \sim \mathcal{N}\left(\log \frac{\rho}{1-\rho}, \sigma^2\right) \quad \text{where } \rho \in (0,1) \tag{26}$$

