# OpenReview forum: "Federated Learning With $L_{0}$ Constraint Via Probabilistic Gates For Sparsity"
_ICLR.cc/2026/Conference — Submitted to ICLR 2026_

### Official Review · Reviewer_x11n · 2025-10-31

**Soundness:** 2
**Presentation:** 2
**Contribution:** 3
**Rating:** 4
**Confidence:** 4

**Summary:**

This paper proposes to enhance Federated Learning (FL) by introducing an L0 constraint on the density of non-zero parameters to enforce model sparsity. This mechanism, implemented using a reparameterization technique with Probabilistic Gates, aims to counteract the issues of overly dense models and poor generalizability arising from unaddressed data sparsity and various forms of heterogeneity common in FL settings. By leveraging techniques like the Binary Concrete Distribution and the Gumbel max trick, the approach allows for flexible model training from either dense or sparse initialization.

**Strengths:**

The paper introduces a novel application of the L0 constraint via Probabilistic Gates into the Federated Learning (FL) framework, specifically targeting model density. This represents a creative combination of established sparsity techniques (like the Binary Concrete Distribution and Gumbel max trick) to address a challenging problem in the unique, distributed FL setting.

**Weaknesses:**

1.While heterogeneity is simulated, the paper needs a more direct and comprehensive comparison against other state-of-the-art FL sparsity or pruning techniques (e.g., FL-adapted L1/L2 regularization, magnitude pruning, or methods based on the Lottery Ticket Hypothesis). This would clearly delineate the unique advantages of the L0 constraint approach.
2.The paper mentions that the parameter p controls the expected number of non-zero parameters. However, a detailed ablation study on the sensitivity of the final performance (accuracy vs. sparsity) to the choice of the sparsity hyperparameter is essential for practical use and is currently missing.

**Questions:**

1.How are the Probabilistic Gate variables specifically treated during the Federated Aggregation step? How does this choice ensure that the sparsity constraint remains satisfied across global model rounds?
2.In highly heterogeneous settings, how are the sparse structures determined by the Probabilistic Gates aggregated across different clients?

---

> ### Author Response · Authors · 2025-12-03
>
> Thank you for your review. 1) we restricted our baselines to reflect communication reductions as regularization based methods gradually reduce communication cost and may be useful for sparse inference but not immediately useful for sparse training. Also, since we are proposing a Lo-based approach, we naturally choose another method based on the Lo pseudo-norm. However, we now have an upper bound for test accuracies from centralized training to compare how close the sparse training in FL got to the upper bound. We hope it alleviates your concerns. 2)The parameter rho is the desired or target density of parameters and this information is passed through the constraint. In a real-world FL scenario, this could be a direct result of the client's device's hardware limitations. We show that FLoPS-PA is efficient for sparse training at very low target densities in both model- and data-sparsity conditions. We also show that a denser model is worse than a sparse model in some cases when there is inherent sparsity in the data or the model, as shown in the experiments section. 3) In gradient aggregation, the gradient with respect to \tilde{theta} and log alpha are communicated, and \tilde{theta} and log alpha are updated at the server, and the probability depends on log alpha. In parameter aggregation, theta and Z are communicated and aggregated at the server. The server infers \tilde{theta} and log alpha using theta = \tilde{theta} \odot Z and the concrete distribution functional relationship, respectively. The server also communicates \tilde{theta} and z back to the client, and the client infers \tilde{theta} and log alpha in the same way, followed by local updating of  \tilde{theta} and log alpha. 4) The sparsity constraint satisfaction happens gradually as the gradients of empirical loss push gates towards the selection of fewer parameters. In addition, clients and servers only send compressed messages (as explained in the algorithm section) in FLoPS-PA, which makes communication sparse, even before and regardless of the time required for exact constraint satisfaction. 5) We aggregate parameters in the same fashion in both homogeneous and heterogeneous conditions. We show that, in both cases, the FLOPS-PA with server-side tuning to mitigate weight divergence empirically converges in the same way.

---

### Official Review · Reviewer_biEp · 2025-10-31

**Soundness:** 3
**Presentation:** 3
**Contribution:** 2
**Rating:** 4
**Confidence:** 3

**Summary:**

The paper presents an FL framework, that enforces $L_0$ constraint on the model parameters to induce sparsity by utilizing probabilistic gates to make the problem tractable. This is motivated by the poor generalization of the dense models in heterogeneous FL settings. The paper further demonstrates that how the $L_0$ constraint is connected to entropy maximization of stochastic gates. They then utilize this insight to derive the $L_0$ constraint for the FL setting. The resulting algorithm called FLoPS allows simultaneous updates of model parameters, gate parameters and a Lagrange multiplier that controls the level of sparsity.

**Strengths:**

1. The key insight in the paper connecting the $L_0$ constraint to entropy maximization of stochastic gates is novel.
2. The utilization of the above insight for deriving the $L_0$ constraint for the FL settings enables a new learning setup.

**Weaknesses:**

1. All experiments in the paper are conducted on linear models, therefore the scalability and soundness of the method on more commonly used non-linear models remain untested.
2. The convergence guarantee or stability conditions of the joint optimization is under-discussed.

**Questions:**

1. Can you provide intuition on the proposed connection between entropy maximization and $L_0$ constraints?
2. Do all clients contribute parameters in every iteration? If not, how are the missing gates handled?
3. How sensitive is the algorithm to the decay and pruning schedule?

---

> ### Author Response · Authors · 2025-12-03
>
> Thank you for appreciating our work. 1)We added experiments with CNN on MNIST data to show our method is applicable in a nonlinear setting as well in FL. 2) We added empirical analysis of FLoPS convergence with random instances of data in the experiments section, and we intend to work on theoretical convergence analysis in the future. We hope new results alleviate your concerns. 3) The pruning in our method is soft, as it is applied using decay r to make the probabilities of a gate being active/inactive more certain than they would be if their indices fell in/out of the top-m theta indices. A higher r pushes probabilities toward certainty more quickly and abruptly. We use a tiny r of 0.001. Also note that this pruning with a higher r(decay) helps gradually reduce the number of active gates and thus the number of non-zero parameters in a setting without compressed communication, which we use and describe in the experiments section. The decay r can also be used at late stages of dense training to prune the model in the last few epochs. We use 0.001 decay and schedule this soft pruning to start from the onset (epoch 0) for FLoPS-PA. 4) Not all the clients participate in training at once; only a fraction of clients participate/available, a source of heterogeneity in FL. However, there are no missing gates because of missing clients, as the gates correspond to parameters, not clients. 5) Intuition: If we do not know which of the parameters are the most essential or true sparsity underneath, then there is uncertainty. One way to determine which parameters are essential is to train every possible combination of parameters, or to add or remove parameters progressively. But this is computationally expensive and intractable for large model sizes. The other way is to quantify the uncertainty. If we can find the unbiased probability distribution that is maximally uncertain to the extent that other signals, like model loss or other limitations, like constraints, allow. This is precisely what an entropy maximization procedure does. An entropy maximization procedure on the states/gates (employed to select or attenuate parameters through reparameterization) with a density constraint, after some approximations, results in a free-energy minimization problem with a constraint that is a continuous, differentiable approximation of the L0 constraint (detailed explanation in the Appendix). The advantage of such an approximation is that the optimization with this constraint is now compatible with gradient-based approaches, allowing us to work with arbitrary and complex loss functions, and also helps build communication compression or reduction mechanisms during training in FL.

---

### Official Review · Reviewer_tYdA · 2025-10-31

**Soundness:** 3
**Presentation:** 2
**Contribution:** 2
**Rating:** 4
**Confidence:** 2

**Summary:**

This paper proposes FLoPS, a federated learning algorithm for learning sparse models with controlled parameter density using probabilistic gates and Hard Concrete relaxation. The authors adapt the L0-constrained optimization framework from Gallego-Posada et al. (2022) to the FL setting, derive the objective from entropy maximization principles, and propose a distributed algorithm based on FedSGD. Experiments on synthetic and real datasets demonstrate superior sparsity recovery and statistical performance compared to magnitude pruning baseline (FedIter-HT) under data and client participation heterogeneity.

**Strengths:**

S1. New prespective on gate-based L0 sparsity to FL: Addresses a gap in the literature where probabilistic gates have not been applied to federated sparse learning.

S1. Extensive experiments across multiple datasets, sparsity levels (0.5%-95%), and heterogeneity conditions (data, client participation).
Consistent improvements over baseline: FLoPS demonstrates better True Discovery Rate and statistical performance than FedIter-HT across all settings.

S3. Achieves target density through constrained optimization rather than tuning regularization coefficients.

S4. Entropy maximization derivation (Section 2.2) provides alternative theoretical perspective on L0 regularization.

**Weaknesses:**

W1. Section 2.2 (entropy derivation) is isolated from the FL application. It derives the centralized formulation but provides no insights for distributed optimization, convergence, or aggregation strategy.

W2.  No theoretical analysis of whether FLoPS converges in heterogeneous FL settings. Does the algorithm converge under non-IID data? How does heterogeneity affect convergence rate? What is the relationship between three learning rates?


W3. The paper applies standard weighted averaging to (\hat{\theta}, \phi) without justification. Why is this optimal? Why not aggregate \theta directly?

W4. How does a consistent global sparsity pattern emerge from heterogeneous local updates when different clients prefer different features? The constraint is only enforced server-side - why is this sufficient?

W5. The paper provide limitted analysis, it requires to have  (i) communication cost comparison, (ii) computational overhead, (iii) ablations on design choices.

W6. Limited baseline comparison: Only compares with FedIter-HT. Missing comparisons with: (i) Lasso-based FL methods (Frandi et al. 2016, Sehic et al. 2022), (ii) Standard FedAvg with post-hoc pruning, (iii) Centralized training (upper bound)

**Questions:**

Q1. Can you provide convergence guarantees for FLoPS under data heterogeneity? What is the convergence rate and how does it depend on heterogeneity level?

Q2. How do you ensure global sparsity pattern emerges from conflicting local preferences?

Q3. What is the total communication cost (rounds × message size)? Does FLoPS converge faster enough to offset the 2× parameter overhead per round?

Q4. refer to weaknesses

---

> ### Author Response · Authors · 2025-12-03
>
> We thank you very much for the detailed review, appreciation, and feedback that will help improve our work. 1) The derivation helps explain the choice for Bernoulli gates for which we use hard concrete relaxation, provides a theoretical framework for sparsity, and further add confidence to the L0 based formulations proposed earlier. Although the derivation looks centralized, We believe it helps view the FL with sparsity as a free-energy minimization problem with a density constraint on states or gates, leading to FLoPS based on SGD and FLoPS - PA : A global free-energy minimizing algorithms , one with aggregation of gradients from clients and the other with aggregation of parameters from clients. We also note that a lagrangian with sum of emperical loss and constraint loss can be written at client level leading to the same composite/global lagrangian. This is a  minimization of local free-energies across clients to minimize global free energy perspective. I hope this explanation clarifies our intent. We added this explanation briefly in the paper, too, in the FL formulation section. 2) We intend to work on theoretical convergence analysis in the future. But we added convergence analysis empirically in the experiments section for both homogeneous and heterogeneous cases for random instances of LR data, and we see that our approach converges faster in both cases and is not affected by heterogeneity. The learning rate for the Lagrange multiplier is fixed at 0.1/|theta| for FLoPS-PA and for other learning rates: we added sensitivity analysis of test performance with respect to the change in learning rate while keeping the other constant in the experiments section. 3) The global sparsity emerges as a result of aggregation of gradients in SGD-based FLoPS and aggregation of locally updated parameters from clients in FLoPS PA 4) We added a FedAvg baseline (densely trained and pruned in the last epochs), centrally trained FLoPS -PA to compare in the experiments section. We did not include regularization-based methods as they lead to gradual sparsification, leading to less reduction in communication cost during training all though if they can be tuned they may still provide inference time sparsity. 5) The constraint is only used on the server side in Fed SGD-based FLoPS as there are no local updates at the client level. In FLoPS-PA (a federated average style parameter aggregation method), which has local SGD steps at each client, the constraint is enforced on the client's side too. 6) In FedSGD-based FLoPS, gradients with respect to \tilde{\theta} and log alpha are averaged, not the parameters themselves. In FLoPS-PA, theta and z are averaged, and \tilde{theta} and log alpha are inferred from their relation to theta and z. We provided this explanation in the algorithms section. 7) We added an explanation / estimation of the communication cost and its comparison in the section with algorithms.

---

### Official Review · Reviewer_zSzV · 2025-11-21

**Soundness:** 3
**Presentation:** 3
**Contribution:** 2
**Rating:** 4
**Confidence:** 3

**Summary:**

The authors seek to tackle the challenges brought by inherent sparsity in data for Federated Learning (FL). To mitigate the risk of building overly dense models and poor generalizability, the authors propose FL with an $L_0$ constraint on the density of non-zero parameters by extending the idea of sparsity in centralized machine learning. The authors reveal the connection between the objective for $L_0$ constrained stochastic minimization and the entropy maximization problem and propose an algorithm based on federated stochastic gradient descent for distributed learning.  The authors conduct experiments on synthetic data as well as MNIST datasets to demonstrate that the proposed method outperforms other methods in both sparsity recovery and statistical performance.

**Strengths:**

I like the deep insights between the objective for $L_0$ constrained stochastic minimization and the entropy maximization problem. Motivated from this connection, the authors are able to propose the federated stochastic gradient descent for distributed learning

**Weaknesses:**

1. While I like the nice insights and connections the authors reveal theoretically, I think there is a big room for improving the experiment section. For example, the largest dataset considered in this paper possibly be MNIST, which is a very small scale dataset. It would lead to questions from readers on the necessity of distributed learning.

**Questions:**

Federated Learning is distributed learning in essence which are often important when dealing with large scale datasets or when devices are hardware-restricted. I would like to see the authors go beyond the small scale size datasets such as MNIST to avoid the challenges on the necessity of distributed learning.

---

> ### Author Response · Authors · 2025-12-03
>
> We are very thankful for the appreciation. As per the suggestion, we included experiments on EMNIST data with 10 times more data and 10 times more clients than our MNIST experiment in the experiments section of the paper. The new results align with all other results. In the future, we would like to expand it to even larger experiments.

---

### Author Response · Authors · 2025-12-04

We sincerely thank all the reviewers for their constructive feedback.

We have added additional experiments, details, explanations, and clarifications to the best of our abilities in the paper and in our official response to improve our paper. We hope our responses and improvements help alleviate any concerns. We request that AC kindly consider our responses and updates to the paper during the assessment.

We are grateful for the time and effort of reviewers and the AC.

---

### Meta-Review · Area_Chair_oMQi · 2026-01-01

**Summary:**

This paper proposes to enforce explicit L_0 density constraints in federated learning via probabilistic gates with continuous relaxation, aiming to induce sparsity and improve generalization under data and client heterogeneity. Motivated by a maximum-entropy perspective, the authors interpret the constrained stochastic optimization problem as entropy maximization over stochastic gates and derive a federated optimization algorithm based on FedSGD / FedAvg-style aggregation (FLoPS and FLoPS-PA). Experimental results on synthetic data and several benchmark datasets (e.g., MNIST, EMNIST, RCV1), together with additional experiments provided in the rebuttal, show that the proposed method can achieve target sparsity levels and often outperforms a magnitude-pruning baseline in terms of sparsity recovery and predictive performance.

The main strength of the work lies in introducing gate-based L_0 sparsification into the federated learning setting and providing a unifying motivation via entropy maximization and free-energy minimization. Explicit control over model density through constraints, rather than tuning regularization coefficients, is appealing and practically relevant. The empirical study explores a range of sparsity levels and heterogeneity scenarios, and the results are generally consistent across settings. The authors also responded constructively to reviewer feedback by adding experiments (e.g., EMNIST, CNNs) and clarifying aspects of communication cost and optimization behavior.

Despite these merits, the overall contribution remains limited. Methodologically, the approach largely adapts existing centralized L_0 probabilistic-gate frameworks to federated learning, with relatively modest conceptual or technical advances specific to the federated setting. The entropy-maximization derivation is largely centralized and remains weakly connected to distributed optimization, offering little insight into aggregation choices, convergence, or stability under non-IID data. Theoretical guarantees are absent, and empirical convergence analyses do not fully compensate for this gap. Experimentally, while some extensions beyond linear models are added, the evaluation still focuses on relatively small-scale benchmarks and lacks comprehensive comparisons with a broader set of state-of-the-art FL sparsity or regularization methods. Communication and computation overhead introduced by probabilistic gates are discussed only at a high level, without sufficiently detailed ablations or trade-off analyses.

In summary, the paper presents an interesting application of L_0 gate-based sparsification to federated learning and offers a coherent empirical study, but it falls short in terms of theoretical depth, experimental breadth, and clear differentiation from prior work. While the rebuttal addresses several reviewer concerns at a surface level, the core limitations remain. Therefore, my overall assessment leans toward reject.

**Reviewer Concerns:**

The rebuttal addresses some of the reviewers’ empirical and clarification-related concerns. In particular, the authors added additional experiments (e.g., EMNIST and a CNN on MNIST), included a few more baselines, and clarified implementation details such as aggregation choices, communication cost estimates, and sensitivity to learning rates. These updates partially alleviate concerns about the narrow experimental scope and missing implementation details.

However, several core concerns remain outstanding. Most importantly, the lack of theoretical analysis or convergence guarantees under data and client heterogeneity was not resolved and is deferred to future work. The entropy-maximization derivation remains largely centralized and still provides limited insight into federated optimization or aggregation behavior. In addition, questions about how a consistent global sparsity pattern emerges under heterogeneous local updates are only addressed qualitatively. Overall, while the rebuttal strengthens the presentation, it does not fully address the main concerns about theoretical depth and federated-specific contributions.

**Reviewer Scores:**

Reviewer zSzV would likely maintain a similar score, possibly with a slight increase, given that their main concern about small-scale experiments was partially addressed by adding EMNIST results, though the overall contribution remains modest.

Reviewer tYdA might remain at the same score, as the rebuttal clarifies several points and adds baselines, but does not address the major concerns regarding theoretical guarantees, convergence under heterogeneity, and aggregation justification.

Reviewer biEp would likely keep their score unchanged. While the added CNN experiment and empirical convergence analysis address some surface-level concerns, the lack of deeper theoretical treatment remains.

Reviewer x11n would also likely maintain their original score. Although additional explanations and comparisons were provided, the limited breadth of baselines and missing detailed ablations on sparsity hyperparameters mean that their main concerns are only partially addressed.

---

### Decision · Program_Chairs · 2026-01-26

Reject